# Grazing Sheep in Organic Vineyards: An *On-Farm* Study about Risk of Chronic Copper Poisoning

**Martin Trouillard** [1],*, **Amélie Lèbre** [1] **and Felix Heckendorn** [2]

1    FiBL France (Research Institute for Organic Agriculture), Pôle Bio, 150 Avenue de Judée, 26400 Eurre, France; amelie.lebre@fibl.org

2    FiBL Switzerland (Research Institute for Organic Agriculture), Ackerstrasse, P.O. Box 5070 Frick, Switzerland; felix.heckendorn@fibl.org

\*    Correspondence: martin.trouillard@fibl.org

**Abstract:** Many winegrowers and sheep breeders are interested in wintertime grazing in vineyards, as an agroecological alternative to mowing or herbicide spraying, and additional supply of forage. Still, strong concern is raised by the use of copper-based fungicides, particularly in organic vineyards, since copper is known to induce chronic toxicosis in sheep. We conducted an *on-farm* study with n = 12 1-year-old Merinos × Mourerous ewes grazing the cover vegetation of vineyard plots during wintertime, in order to check whether this agricultural practice might be harmful to sheep. Our results indicate that most copper found in the cover vegetation originates from fungicide spraying *versus* plant uptake from the soil, and that rain-induced washing-off and plant growth-triggered dilution of copper are crucial to reach close-to-safe grazing conditions. Furthermore, we found that while sheep remained globally healthy during the 2 months of the experimental period, the plasma activity of Glutamate Dehydrogenase increased by 17.3 ± 3.0 U/L upon vineyard grazing ($p < 0.001$), reflecting liver storage of copper. We also discovered that the dynamics of molybdenum in sheep plasma are strongly affected by exposure to copper, suggesting a possible adaptation mechanism. Overall, our results suggest that winter grazing of sheep in organic vineyards is reasonably safe, but that care should be taken about grazing period duration. More research should be conducted with respect to long-term copper accumulation, spring and summer grazing, and possible protective mechanisms against copper chronic poisoning.

**Keywords:** copper; sheep; vineyards; organic winegrowing; molybdenum; agroforestry; crop-livestock integration

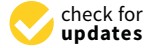

## 1. Introduction

Animal grazing in perennial cultures is an ancient practice that has been maintained in some modern agricultural contexts [1,2]. In particular, sheep grazing in vineyards during vine dormancy can bring the benefit of free pasture resources in winter, and may contribute to weed management in the vineyard without recourse to mechanical or chemical means, thus allowing significant savings [3]. Additionally, sheep grazing seems to improve soil fertility and carbon sequestration in vineyards [4], and might also contribute to the regulation of diseases and pests in orchards [5]. Sheep are even sometimes used as cultivation auxiliaries, plucking vine leaves at no cost and with higher efficiency than humans do [3]. For all these reasons, this crop–husbandry association that had been somehow abandoned as a consequence of agricultural specialization, is regaining much interest among farmers, and can be viewed as a potentially major agroecological leverage towards more sustainable production of perennial crops [6].

However, risks of sheep Copper Chronic Poisoning (CCP) have long been reported, and may hinder this practice [7,8]. Indeed, copper-based fungicides have been and still are widely used in vineyards, and on the other hand, sheep are very prone to copper intoxication. While copper is a component of many pesticides in conventional agriculture,

it is almost essential in organic agriculture, where it stands as one of the only efficient agents for the control of downy mildew and other crop diseases [9,10]. The allowed amount of copper to be annually sprayed on organic vineyard plots has recently been reduced to 4 kg Cu/ha/year (averaged over 7 years) in the EU, while up to 80 kg Cu/ha/year were used in the early 20th century [9]. Since Cu mobility in soils is low, it has been accumulating in plots with a long vineyard history, reaching levels of 200–400 ppm and more in certain areas, and consequently triggering long-term negative effects on the soil micro- and macro-fauna [11,12]. As a consequence, cover vegetation growing on this kind of soil has been shown to harbor abnormally high Cu concentrations higher than 20 mg Cu/kg dry matter (DM) [12,13].

CCP in sheep is a long-term phenomenon that can occur upon continuous ingestion of copper at levels as low as 15–17 mg Cu/kg diet DM [14,15]. During exposure, Cu is asymptomatically stored in the liver (and to a lesser extent, in the kidneys) of the animals, where it can reach concentrations of 1000 to 4000 mg Cu/kg of liver DM [15–17]. When the concentration of Cu in the liver is high, it can be suddenly released upon any slightly stressful event, causing massive damage to erythrocytes. In the absence of treatment, such a "hemolytic crisis" leads to the death of the animals within 1–2 days, although in some cases, mortality has rather been associated to global liver failure [15]. CCP can be a concern for ruminants in general, but sheep are particularly prone to it since they seem to have a comparatively inefficient liver excretion mechanism [14,18]. Risk of CCP can be increased if additional hepatotoxic factors are involved, such as consumption of *Heliotropium europaeum* [19] or *Senecio jacobaea* [20].

Since CCP remains subclinical in the Cu storage phase, the best way to assess the probability of sheep to endure a hemolytic crisis is to measure the Cu content of their liver and/or kidneys, through collection of organ samples or biopsies. However, as Cu storage in the liver is correlated with progressive necrosis of hepatocytes, the concentration of specific liver enzymes in the plasma can be used as a noninvasive indicator of CCP status as well [16,21,22].

Cu absorption and liver storage is subjected to antagonism towards other mineral elements. Among them, molybdenum and sulfur are of major importance, mainly under the form of tri- and tetra-thiomolybdate. These molecules create complexes with Cu, preventing its absorption and liver storage [15,23], and promoting its release from hepatocytes [24,25]—thereby potentially leading to symptoms of Cu deficiency. Zinc and iron may also contribute to Cu antagonism, albeit at much higher doses than Mo [17,26].

Today, the few existing recommendations about maximal quantities of copper spraying in order to keep winegrowing compatible with sheep grazing are not supported by experimental data, and knowledge about the occurrence of CCP in vineyard conditions is still lacking. Moreover, the relative importance of annually sprayed copper versus soil-accumulated copper uptake by plants is still uncertain. In an attempt to fill these knowledge gaps, the present *on-farm* study thus explored for the first time the possible advent of CCP in sheep grazing in organic vineyards during winter, focusing on the contents and origin of Cu in the vegetation cover of the vineyards, and comparing non-invasive markers of the CCP status of exposed animals.

## 2. Materials and Methods

### 2.1. Experimental Setup

The first part of the experiment was designed to provide information about Cu dynamics in the cover vegetation and soil. To do so, the cover vegetation of two vineyard plots in the area of the Clairette de Die wine production in Drôme, France, were sampled beginning 8 days after the last application of Cu-based fungicide (Figure 1, green arrows), for analysis of Cu content. Similar samplings were conducted at defined intervals thereafter, with the aim of following the decrease of Cu content in the vegetation over time.

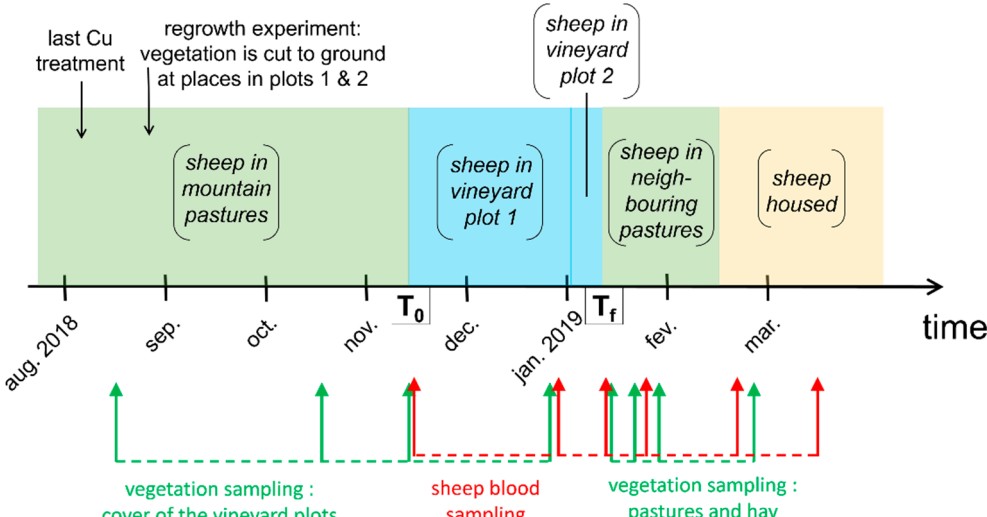

**Figure 1.** Experimental setup and timeline. Samples of the cover vegetation of the vineyard plots (lower part, green arrows) were collected repeatadly after the end of treatments with Cu-based fungicides and when sheep were introduced to new plots. At $T_0$, sheep were allowed to graze in two vineyard plots consecutively, ending at $T_f$ (blue panels); they were then led to surrounding pastures, and eventually housed and fed on hay (green and yellow panels, respectively). Blood samples (lower part, red arrows) were collected at $T_0$ and at defined intervals during and after vineyard grazing.

At four randomly chosen places of both vineyard plots, vegetation was cut to ground on 30 August 2018 using a string trimmer, in order to monitor Cu content of a regrown vegetation that never received any Cu application. Additionally, soil samples were collected in both plots to better understand the link between copper in the plants and soil.

Then, starting on 15 November 2018 ($T_0$), n = 12 sheep were allowed to graze successively in vineyard plots 1 (until 2 January 2019) and 2 (until 10 January 2019: $T_f = T_0 + 56$ days) (Figure 1, blue panels). Grazing duration was particularly short in plot 2, because vegetation had been cut to ground under the vine rows at harvest time, and had almost not regrown; and because 1/3 of the field had not been harvested, impeding the introduction of sheep in this area in order to prevent them from eating rotten grapes—and therefore diminishing the grazed area of plot 2 to 1.23 ha (see Section 2.2)

After the vineyard grazing period, the sheep of the experiment were gathered with the rest of the farmer's flock, and allowed to graze various pastures of the area, including cover crops, permanent meadows, and underwood pastures (Figure 1, green arrows). On 15 February 2019 (Tf + 36 days), sheep were housed and fed on good quality hay without any additional concentrate (Figure 1, yellow panel).

Blood samples were collected from all sheep at $T_0$, $T_f$, and at the time of plot changing ($T_0 + 42$ days); as well as after the end of the grazing period, at $T_f + 12$, 39, and 63 days (Figure 1, red arrows). Vegetation of both plots was sampled shortly before sheep arrival, for determination of Cu, Mo and S content. Vegetation of the plots grazed by the sheep after their presence in vineyards, as well as hay samples were analyzed in the same way.

## 2.2. Sheep and Plot Description

Sheep were crossbred Merino × Mourerous, 1 year-old ewe-lambs, that had spent the summer season on nearby mountain pastures. Grazing in vineyard plots was conducted in a dynamic manner, successively allowing sheep to access small areas of the plots when the vegetation had been satisfyingly consumed.

Plots 1 and 2, of 1.94 ha and 1.84 ha, respectively, were separated by a distance of 850 m, and were essentially composed of the same type of soils (Rendzic Leptosol). Both were planted with vines of variety Clairette, at 2.40 m width between two rows. In plot 1, every other inter-row was tilled, while the other was composed of a regularly mown

spontaneous cover; vegetation of the inter-rows was mown at the end of August 2018 to facilitate harvesting, but no mowing was performed under the vine rows before or after harvesting. In plot 2 all the inter-rows were covered with a sown mixture of *Festuca rubra* and *Lolium perenne,* and mown before harvest on 18 August 2018. The area covered by vine rows, covered with ground vegetation or not, was estimated to be 0.8 m wide in both plots, thus representing 1/3 of the plot surface.

Plot 1 was under organic agriculture certification, and plot 2 was managed with limited use of synthetic pesticides. Cu-based fungicides were sprayed in both plots soon after budding (4 May 2018 in both plots) and approximately until veraison (16 July 2018 and 6 August 2018 in plots 1 and 2, respectively). Bordeaux mixture ($CuSO_4$) and copper hydroxide ($Cu(OH)_2$) were the only forms of copper applied in 2018, generally in mixture with dispersive agents. Total doses of applied Cu amounted to 3.29 and 5.93 kg Cu/ha in 2018 in plots 1 and 2, respectively. Sulfur-based fungicides were also sprayed on both vineyards, amounting to 41.0 and 68.3 kg S/ha in plots 1 and 2, respectively. These above-average quantities of Cu and S sprayings can be explained by the very rainy spring of that year, which was particularly favorable to the development of downy and powdery mildew.

### 2.3. Sampling and Analysis

In order to perform vegetation sampling, plots were divided in 3 subplots. These were subsequently sampled by randomly applying a 0.25 m$^2$ square 3 times in each subplot, harvesting all the cover vegetation delimited by the square, and pooling the collected plants together. Plants were cut using stainless steel scissors taking particular care to avoid contamination by soil particles, stored in plastic bags, and frozen at −18 °C. Then, a representative fraction of the samples was used for quantification of Mo, Cu, and S by ICP-OES after mineralization by fluorhydric acid (Mo, Cu) or nitro-hydrochloric acid (S). Presented results are the mean of the values obtained for each subplot.

In the regrowth experiment, (see Section 2.1) vegetation was assayed in the same way as described above.

Soil surface samples were collected with a metallic spade in the first 20 cm, similarly pooling 3 samples in each subplot, and averaging values from the subplots. Samples were kept at −18 °C until standard agronomic analysis were performed. Soil EDTA-extractable Cu was measured by ICP-OES after ammonium acetate extraction in presence of EDTA according to the procedure described in the French norm NF X31-120, while total soil Cu was determined after mineralization by nitro-chloric acid.

Sheep blood samples were collected from the jugular vein using heparinized tubes, and plasma was stored at −18 °C after centrifugation at 2500 rpm during 10 min, until Cu, Mo, Fe concentrations, and activity of hepatic enzymes were determined by ICP-OES.

Activity of Gamma-Glutamyl Transferase (GGT) and Aspartate Aminotransferase (AST) was measured spectrophotometrically from the plasma, according to the IFCC standards [27], with the addition of pyridoxal phosphate in the case of AST. Glutamate Dehydrogenase (GLDH) activity was determined by monitoring the oxidation rate of NADH at 340 nm. All spectrophotometric assays were performed on a Beckmann AU480 spectrophotometer (Beckmann-Coulter, Brea, CA, USA).

Tri-chloroacetic acid (TCA) extraction was performed according to the method published by Koontz [28]: 750 μL of plasma were mixed and vortexed with an equal volume of TCA solution (10% *w/w*), and left to precipitate during 30 min at 4 °C. After 10 min centrifugation at 10,000× *g*, the pellet was resuspended in 500 μL TCA, and the operation repeated. Supernatants were kept at −18 °C for later determination of Cu and Mo contents by ICP-OES.

Parasitological assays were conducted to assess the absence of infection by the liver fluke (*Fasciola hepatica*), which could trigger hepatic lesions and subsequent enzyme release, similar to those caused by CCP. Sheep feces were sampled, and 5 g of each sample was ground in the presence of water, and sieved at 250 μm. Then, the filtrate was allowed to settle for 30 min, after which the supernatant was removed, and the sediment was

subjected to another two cycles of washing/10 min sedimentation. The obtained sediment was colored with methylene blue, and observed under the microscope ($\times$40) for checking the possible presence of undyed, yellowish *Fasciola hepatica* eggs.

Meteorological data were acquired by MeteoFrance at the station of Beaufort sur Gervanne (coordinates: 44.778, 5.139), located at 2.4 and 3.2 km of plots 1 and 2, respectively.

### 2.4. Data Treatment and Calculations

The formulas proposed by Suttle were used for determining the way Mo and S can synergistically modulate Cu absorption from grass and hay, respectively [17]:

$$A = 5.7 - 1.3 \cdot 10^{-3} * [S] - 2.785 * \log[Mo] + 2.27 \cdot 10^{-4} * [Mo] * [S] \tag{1}$$

$$A = 8.9 - 0.7 * \ln[Mo] + 2.61 \cdot 10^{-3} * \ln[S] \tag{2}$$

where $A$ is the percentage of consumed Cu that is absorbed by the animals.

Using these equations, theoretically absorbed Cu from the cover vegetation in the vineyards of the study was calculated, and converted into amounts of Cu ingested per ewe and per day, on the basis of a daily grass intake of 1.5 kg DM. These in turn can be related to the liver mass of young ewes with 30 kg live weight, using a DM for liver of 0.48% of body mass [29]. 'Safe' duration for grazing in these plots was defined as the time needed to reach a level of 1000 mg Cu/kg liver DM, considered as a threshold upon which hemolytic crisis is likely to happen [17]—assuming that during this period, Cu release from the liver is negligible. This duration was therefore calculated as follows:

$$t = \frac{1000 * 0.0048 * 30}{1.5 * \frac{A}{100} * [Cu]} = \frac{9600}{A * [Cu]} \tag{3}$$

When applicable, a factor 1/3 was used to represent the contribution of rows to the overall risk, while a factor 2/3 was applied to inter-rows, in agreement with the relative plot area occupied by these zones (see Section 2.2).

Humann-Ziehank et al., observed a good correlation between activity of GLDH and Cu content in the liver for two sheep breeds [16]. In our experiments, we therefore calculated theoretical Cu liver contents on the basis of their formula, using a liver dry matter ratio of 0.263:

$$[Cu]_{liver\ DM} = \frac{1}{0.263} * 10^{\sqrt{\frac{\log(GLDH) - 0.625}{0.145}}} \tag{4}$$

When applicable, Student's two sample *t*-tests were used to determine the statistical significance of datasets. Normality of the datasets' distribution was previously checked using Shapiro–Wilk test. All data were treated and analyzed using Origin 2020 software (OriginLab Corporation, Northampton, MA, USA); algorithms used for the calculation of statistical significance can be found on the Origin software webpage: https://www.originlab.com/doc/Origin-Help/tTest-TwoSample-Algorithm (accessed on 9 November 2021).

### 3. Results and Discussion

### 3.1. Cu in the Vegetation Cover of Vineyard Plots

3.1.1. Dynamics of Cu in the Plants and Soil

Strong precipitations (59.5 and 51.5 mm for plots 1 and 2, respectively: Figure 2, blue line) occurred between the last Cu-based fungicide treatment of the season and the sampling of cover vegetation. Extrapolating experimental models of Cu washing-off from other kinds of leaves, it can be expected that most of the washable Cu had been removed from the cover vegetation by these rain events [30–32]. Despite this, the measured total Cu contents in the cover vegetation on 14 August 2018 ranged from 42.8 to 64.7 mg Cu/kg DM in the vine inter-rows of plots 1 and 2, respectively, and reached 225 mg Cu/kg DM under the vine rows of plot 1 (Figure 2, green lines and symbols), where unintentional spraying of Cu was probably maximal.

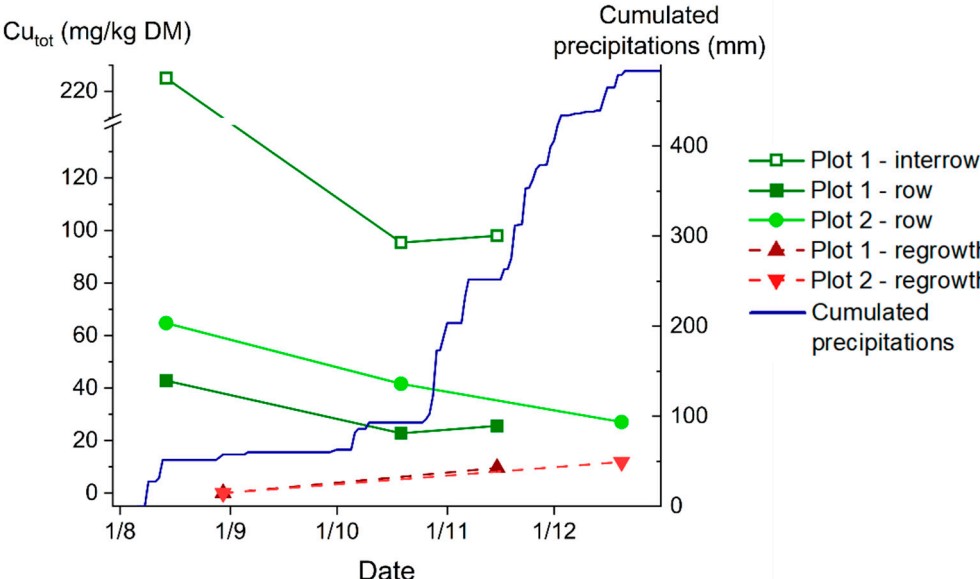

**Figure 2.** Total Cu content of the vegetation cover in vineyard plots over time. Samplings were discontinued when sheep entered the vineyard plots. Total Cu content of the regrown vegetation is plotted at the time of sheep introduction, and is considered 0 at the time of cutting down.

These values for Cu content largely exceed the proposed thresholds for CCP risk in sheep [14,17], and it is not expected that the remaining Cu will be washed off by further rain events. Still, it can be seen on Figure 2 (green lines and symbols) that the total Cu content in the cover vegetation decreases by at least 30% and up to 50%, during the following 3 to 4 months that precede grazing in the vineyard plots. This evolution must therefore be explained by the dilution of Cu under the influence of plant growth.

On the other hand, plant Cu uptake through their root system could be expected to balance this dilution effect. In our experiment, cover vegetation that had been cut to ground and never received any Cu spraying harbored Cu contents of $9.2 \pm 0.6$ and $11.8 \pm 1.0$ mg Cu/kg DM after 77 and 112 days, respectively, in plots 1 and 2 (Figure 2, respectively dark and light blue). Though higher than the Cu content of the vegetation in uncontaminated surrounding pastures (Table 1), these concentrations are low to moderate and consistent with the level of Cu in the soils of the vineyard plots used in this study, ranging from $13.1 \pm 6.1$ (plot 2) to $32.0 \pm 5.8$ mg Cu/kg soil DM (Table 2). This in turn is consistent with a relatively short winegrowing history of 20–30 years in the concerned plots, and with the soil parameters being globally unfavorable to Cu transfer to plants (relatively high pH, CEC, and organic matter content—Table 2) [12,33]. Additionally, growing plants tend to take up more Cu than older ones [34], which probably leads to an overestimation of the contribution of Cu uptake in the total amount of Cu in the vegetation cover. Our findings thus suggest that under these conditions, the contribution of soil Cu to the possible advent of CCP in sheep can be considered as negligible. Still, attention should be paid to this factor if sheep are to graze heavily Cu-loaded plots, especially on acidic soils poor on organic matter.

**Table 1.** Cu, Mo, and S contents of the vegetation cover in the vineyard plots of the experiment, in the surrounding open and underwood pastures used for grazing sheep afterwards, and in the hay fed after the sheep were housed. Values are means of n = 3 measurements ± standard deviations.

| | Cu | +/− | Mo | +/− | S | +/− | Cu/Mo | Absorbable Cu | Calculated Pasture Time |
|---|---|---|---|---|---|---|---|---|---|
| | mg/kg DM | | mg/kg DM | | mg/kg DM | | | % | Days |
| Plot 1 (rows) | 98.0 | 17.5 | 0.46 | 0.02 | 1560 | 245 | 214.5 | 6.0 | 16 |
| Plot 1 (inter-rows) | 27.9 | 0.8 | 0.58 | 0.01 | 2600 | 184 | 48.5 | 4.2 | 82 |
| Plot 2 (inter-rows) | 36.8 | 15.9 | 0.46 | 0.08 | 2775 | 318 | 80.8 | 4.6 | 57 |
| Pastures | 5.1 | 0.4 | 2.44 | 1.49 | 2430 | 834 | 2.1 | 1.4 | 1335 |
| Underwood pastures | 4.5 | 1.3 | 0.82 | 0.38 | 1130 | 77 | 5.5 | 5.0 | 430 |
| Hay | 1.0 | 0.3 | 5.91 | 2.38 | 1123 | 375 | 0.2 | 7.4 | 1259 |

**Table 2.** Characteristics of the soils of vineyard plots 1 and 2 used in the study. OM: Organic Matter content, CEC: Cation Exchange Capacity. For Cu and Cu EDTA, samples were collected from the soil of both the inter-rows and under the vine rows; values are means of n = 4 ± standard deviations.

| | pH | OM | CEC | Cu | +/− | Cu EDTA | +/− |
|---|---|---|---|---|---|---|---|
| | | g/kg | méq/kg | mg/kg | | mg/kg | |
| Plot 1 | 7.94 | 29.1 | 161 | 32.0 | 5.8 | 16.6 | 4.1 |
| Plot 2 | 7.83 | 31.4 | 117 | 13.1 | 6.1 | 6.4 | 4.2 |

Taken altogether, our results advocate for the need for significant (>20 mm) precipitation after spraying Cu-based fungicides, before sheep are to be introduced in vineyard plots; and for letting grass grow as much as possible in order to benefit from a dilution effect. Given the high Cu values revealed by our measurements in the cover vegetation, strong concern is raised about the compatibility of year-round presence of sheep in organic vineyards or orchards, and leaf-plucking of vines by sheep, with the health status of the latter towards CCP.

3.1.2. Antagonism and Risk of CCP

Since absorption of Cu can be prevented by its association with Mo and S in the rumen of sheep, these elements were determined in the cover vegetation of vineyard plots 1 and 2 at the onset of sheep grazing, as well as in surrounding pastures (Table 1). Surprisingly, the sulfur content in the cover vegetation was not considerably higher in vineyard plots, despite the intensive use of sulfur in organic viticulture. Molybdenum was generally found in lower concentration (<0.7 mg/kg DM) in vineyard plots than in other areas.

To take into account antagonism effects, it has been suggested that the Cu:Mo ratio of the feed might reflect the CCP risk with more accuracy than the Cu content itself, and that this ratio should be kept under a limit of 20 [17]. In the vineyards grazed during the experiment Table 1 shows that this threshold was largely exceeded, sometimes by a factor of 10 (plot 1, under the vine rows).

Taking into account the contents of both Mo and S in the vegetation covers, we calculated that sheep should reach the end of the "safe pasture time" (see Section 2.4) after grazing the vineyards of the study after respectively 60 and 57 days in plots 1 and 2 (Table 1)—while CCP is expected to never occur in the surrounding woods and pastures, or upon feeding on hay ('safe time' > 1 to 3 years). It should be kept in mind that these calculations are based on several assumptions, and therefore only constitute a proxy for estimating the risk of CCP for sheep grazing in vineyards (see Section 2.4). Still, on the basis of these calculations, the theoretical amounts of Cu effectively absorbed by sheep can be viewed as worrying if they are continuously grazing in the vineyards during 2 months or more—which are the conditions of our experiment.

### 3.2. Monitoring the CCP Status of Sheep by Blood Analysis

#### 3.2.1. Hepatic Enzymes

GLDH activity in the plasma increased when sheep grazed in vineyard plots (Figure 3a). As a matter of fact, GLDH values at $T_f$ and $T_f$ + 12 days were significantly different from the initial values ($p < 0.05$ and $p < 0.001$, respectively). The maximum value for average GLDH was reached after the end of Cu exposure (at $T_f$ + 12 days), and corresponds to a theoretical levels of liver Cu of 382 mg Cu/kg DM liver, which should discard any risk of hemolytic crisis [14,17].

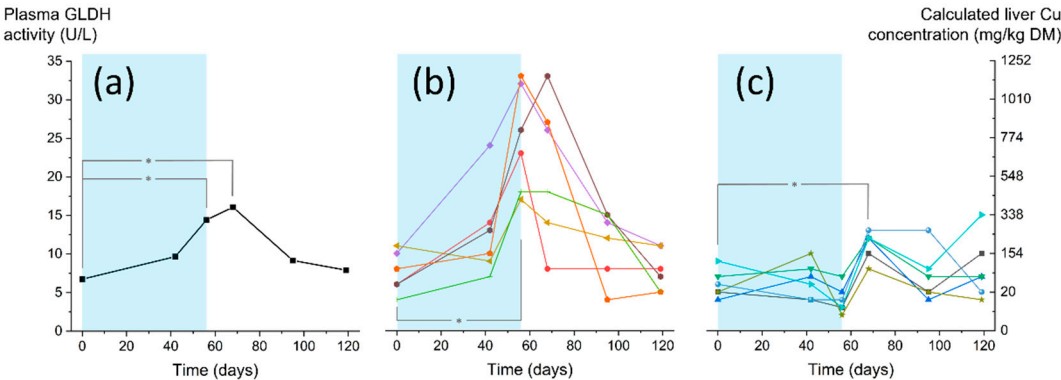

**Figure 3.** Plasma GLDH activity (left axis) and calculated Cu concentration in liver (right axis) of sheep grazing in vineyards (light blue panel = time of presence in the vineyard plots). (**a**) Average curve of all n = 12 sheep; (**b**) Curves for the n = 6 sheep harboring the highest values of GLDH activity and (**c**) for the remaining n = 6 sheep. Grey lines and asterisks indicate significant differences, described in the text.

However, individual GLDH curves show that the evolution of GLDH in the plasma was subjected to wide inter-individual variations. Indeed, half of the animals harbored values higher than 15 and up to 33 U/L at $T_f$ or $T_f$ + 12 days (Figure 3b), while the others never exceeded 15 U/L throughout the experiment (Figure 3c). In the 'high' group, GLDH values at $T_f$ were significantly different from the initial values ($p < 0.001$, with a calculated difference of $17.3 \pm 3.0$ U/L), whereas they were not in the 'low' group ($p > 0.05$). Still, even the 'low' group underwent a slight but significant increase of its GLDH activity at $T_f$ + 12 days, reaching $5.2 \pm 1.0$ U/L higher values than at $T_0$ ($p < 0.001$). For the 3 animals that harbored the highest GLDH values, the corresponding liver Cu concentrations are estimated to be higher than 1100 mg Cu/kg DM liver (Figure 3b), a range in which CCP can be considered as probable, and where a hemolytic crisis is likely to happen [15,17].

Such a strong variability amongst sheep towards CCP has been reported before [16,35], but remains largely unexplained to our knowledge. Occurrence of the maximal GLDH plasma activity after the end of Cu exposure had also been documented in previous studies [16]. Both the overall slight increase of GLDH activity, and large inter-individual differences, were observed in previous studies of our group under similar experimental conditions (data not shown).

Surprisingly, the initial average AST activity was found to be abnormally high ($157 \pm 26$ U/L) when compared to literature data [17]. AST values were significantly lower at $T_0$ + 42 days and $T_f$ + 12 days than at $T_0$ (differences were respectively $-34 \pm 11$ U/L and $-29 \pm 9$ U/L, $p < 0.005$), but otherwise remained globally stable throughout the course of the experiment (Figure 4a). The observed decrease of AST activities upon Cu exposure is hard to understand, since high AST values are supposed to reflect liver damage caused by CCP. Noticeably, the 6 ewes of the 'high' GLDH group showed significantly higher overall AST values than the others ($p < 0.01$).

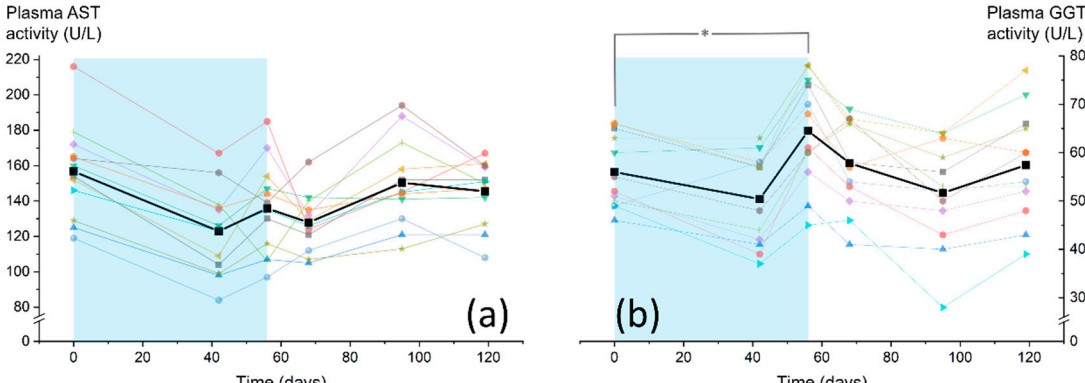

**Figure 4.** (**a**) Activity of AST and (**b**) GGT in the plasma of sheep grazing in Cu-treated vineyard plots (light blue panel = time of presence in the vineyards). Bold black curves and symbols are the mean of the n = 12 individual values shown as light, colored lines and symbols (same color codes for individual sheep as in Figure 3). Grey line and asterisk indicate a significant difference, described in the text.

Meanwhile, the measured GGT activities slightly but significantly increased at $T_f$ in all samples, compared to their values at $T_0$ (Figure 4b; $GGT(T_f) = GGT(T_0) + 8.5 \pm 3.9$ U/L; $p < 0.05$), similarly to the observation made on GLDH, but always stayed under the threshold values for CCP [17].

The largest effects of vineyard grazing on plasma activity of hepatic enzymes was observed for GLDH, which therefore appears as the most reliable indicator for non-invasively monitoring ongoing CCP, in agreement with previous works [16]. Based on our data, we believe that exposing sheep to Cu in vineyard plots induced slight liver Cu storage, damage to hepatocytes, and therefore the beginning of a risk of a hemolytic crisis. However, this phenomenon happened to a smaller extent than expected on the basis of our measurements of mineral concentrations in the cover vegetation of the vineyard plots (see Section 3.1.2). Interestingly, while the induced effects were hardly sensible in most sheep, up to 1/4 of the animals reached theoretical levels of liver Cu that could be sufficient to trigger hemolytic crisis. Further research should notably be designed to provide better understanding of these inter-individual variations.

### 3.2.2. Concentration of Mineral Elements in the Plasma

The plasma samples used for testing the activity of hepatic enzymes were also assayed for estimating the abundance of mineral elements, including Cu, Mo, Fe, and Zn. While the two latter did not show significant variations (data not shown), the Mo concentration in plasma underwent important and reproducible changes upon introduction of the sheep in the Cu-treated vineyard plots (Figure 5a). Indeed, Mo quickly disappeared from the plasma of sheep, uniformly reaching the limit of detection (10 µg/L), until the end of the presence of the sheep in the vineyards. Then, this parameter strongly increased, to levels much higher than the values at the beginning of the experiment (240.8 ± 71.0 vs. 65.8 ± 7.7 µg/L initially). In contrast with GLDH activity, the concentration of Mo in the plasma was not subject to wide inter-individual variations.

This drastic decrease of the Mo plasma concentration can be interpreted as a mechanism towards prevention of Cu storage in the liver of sheep, which to our knowledge has not been documented before. When Mo is depleted from the plasma (and from possible other body reservoirs), Cu liver storage is likely to begin. In this view, the discrepancy between the fairly high potential of the vegetation cover to trigger CCP in sheep, and mild effects as reflected by the release of hepatic enzymes in the plasma, could be explained by a relatively high initial Mo level of the animals involved in our experiment—which would have exerted a protective effect in the first phase of Cu exposure. The high Mo content of the hay habitually fed to the sheep (Table 1) advocates in favor of this hypothesis, since the animals might have stored Mo from previously fed forage.

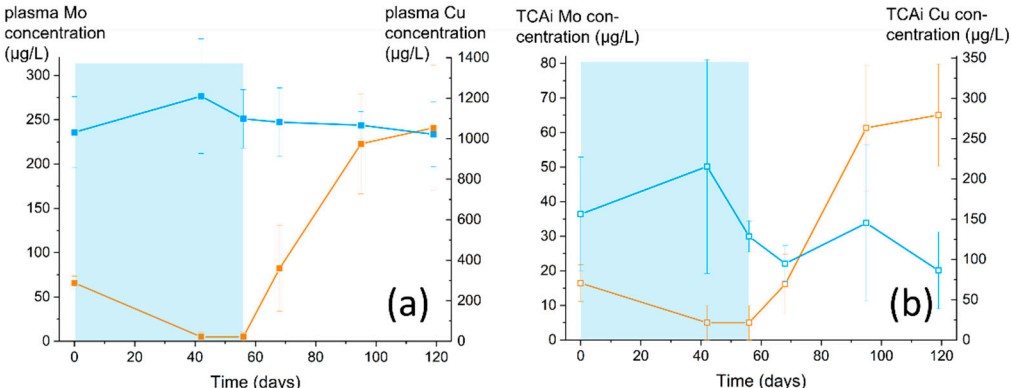

**Figure 5.** (**a**) Mo and Cu (respectively orange and blue lines and closed symbols) concentrations over time in the plasma of sheep grazing in vineyards (light blue panel = time of presence in the vineyard plots); (**b**) Mo and Cu concentrations in the TCAi fraction of the same plasma samples. Values are means of n = 6 individual measurements, error bars represent standard deviations. Values of Mo concentration under the limit of detection of 10 μg/L are arbitrarily set at 5 ± 5 μg/L.

"Normal" Mo plasma concentrations of sheep have been described at around 50 μg/L, but can rise to more than 2 mg/L if sulfur is present at low quantities [36,37]. Still, some sheep have been found to come up with almost no Mo in their plasma at the beginning of experiments in orchards (Trouillard et al., ongoing research program). Therefore, Mo determination in plasma may not be sufficient: the concentration of this mineral in "storage" organs and tissues (liver, kidney [38], but also bones [39]) should be checked prior to Cu exposure, in order to investigate more precisely the potential effect of Mo supplies against CCP.

On the other hand, we have hypothesized that the 'rebound effect' of Mo concentration in the plasma after the end of grazing in the vineyards could be linked to the persistence of plasma thiomolybdate–Cu complexes. Indeed, Mo administration (under the form of tri- or tetra-thiomolybdates) has been proven to increase Cu excretion from systemic sources including liver storage, but also to induce Cu binding and cause transitory retention of a thiomolybdate–Cu complex in the TCA-insoluble (TCAi) fraction of the plasma [40,41]. In this view, the high Mo levels in our experiment may be interpreted as an evidence for ongoing release of Cu from the liver of sheep, consecutive to the end of Cu exposure.

To check this hypothesis, we measured the levels of Cu and Mo in the TCAi fraction of the plasma of sheep subjected to grazing in the vineyards (Figure 5b). Our results indicate no evidence of Cu retention, since the Mo concentration in the TCAi fraction was proportional at all time to that in the whole plasma, with a regression factor of 3.72 ± 0.15 (R = 0.99543); moreover, Cu concentration in the TCAi fraction was not significantly increased after sheep had been exposed to Cu-rich vegetation.

Therefore, the best remaining possible explanation for the strong rise in plasma Mo after sheep had grazed in vineyards should be a spectacular increase of their Mo uptake capacity. Mo absorption by sheep is believed to be an active carrier-mediated process, probably sharing its transporter—and therefore, competing—with sulfate [42]. However, a specific Mo transporter has been identified recently in eukaryotes, which is over-expressed in conditions of Mo deficiency [43]. Although such a molecular mechanism can hardly be evidenced in *on farm* studies, it may play a crucial role in the management of sheep herds in vineyards. Upon Cu exposure, sheep might indeed develop an adaptive up-regulation of Mo metabolism, which could play a protective role if they are successively driven in and out of Cu-rich plots—which fortunately corresponds to the usual practice.

## 4. Conclusions

For the first time, an extensive study of the risk for sheep to develop CCP while grazing in Cu-treated vineyards has been conducted. Our results indicate that direct drifting from spraying of Cu-based fungicides is the main source of CCP risk, and that it should be so in

other conditions, as long as the soil characteristics are not massively in favor of Cu transfer to plants. Advice should be given to delay as much as possible the introduction of sheep in vineyard plots after harvest, since Cu concentrations strongly decrease in autumn under the influence of rain-induced washing-off, and dilution provoked by plant growth.

Overall, the risk towards sheep mortality appears to be low if the grazing duration is not too long (ca. 2 months). However, 'sheep breed' should be taken into account, since this factor can determine wide variations of sensitivity to CCP. Additionally, for unclear reasons, it seems that some sheep might suffer adverse effects due to Cu overload, when other animals of the same group may be out of risk. Eventually, we cannot exclude that the sheep used in our study benefited from the protective effect of an important Mo load prior to Cu exposure, which might not be the case in other flocks.

Therefore, grazing sheep in Cu-treated vineyard plots appears to be reasonably safe, but care needs to be taken, abnormal behaviors, diseases or mortalities should be diagnosed, and research needs to be extended in order to guarantee full safety of this agro-ecological practice. In this view, work still needs to be done in order to better understand inter-individual variations; to further explore the role of Mo uptake by sheep; and to decipher the possible long-term cumulative liver Cu storage. Situations of higher Cu exposure, such as leaf-plucking of vines by sheep, or grazing during the spring and summer seasons, would thus be systems of high interest for future studies.

**Author Contributions:** Conceptualization, experimental design, data collection, M.T., A.L. and F.H.; material resources, A.L.; Data analysis and graphing, M.T.; Writing—first draft of the manuscript, M.T.; writing—review and editing, M.T. and F.H.; supervision, F.H. All authors have read and agreed to the published version of the manuscript.

**Funding:** This research was conducted in the frame of the project "Brebis et Clairette de Die, pâturer pour mieux désherber", which was funded by the French Water Agency (Agence de l'Eau Rhône-Méditerranée), grant No. 2017 6394. Partners of the project were the Communauté de Communes du Val de Drôme, Syndicat de la Clairette et des Vins du Diois, and Fédération Départementale Ovine de la Drôme.

**Institutional Review Board Statement:** Ethical review and approval were waived for this study, since all the samplings were performed under the supervision of the veterinarian in charge of the flock, in the aim of preserving animals' health.

**Informed Consent Statement:** Not applicable.

**Data Availability Statement:** The datasets generated during and/or analyzed during the current study are available from the corresponding author on reasonable request.

**Acknowledgments:** The authors are particularly grateful to the farmers that welcome the study, namely Serge Krier, Fabien Lombard and Nicolas Peccoz. Many thanks are also addressed to the project partners Elise Chevalier, Sophie Ferreyra, Nathalie Groulard and Louise Riffard. Additionally, the authors would like to warmly thank Hervé Pouliquen for his useful precisions about copper toxicity, and Arnaud Dufils for proof-reading the manuscript and providing meaningful comments.

**Conflicts of Interest:** The authors declare no conflict of interest.

**Ethics Approval:** All blood samplings were performed by and under the supervision of the veterinarian in charge of the flock. Animal welfare guidelines were fully respected.

## Abbreviations

CCP: Chronic copper poisoning; DM: Dry Matter; GGT: Gamma-Glutamyl Transferase; AST: Aspartate Aminotransferase; GLDH: Glutamate Dehydrogenase; TCA: Trichloroacetic acid; OM: Organic matter; CEC: Cation exchange capacity.

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
