# Peer review of "Grazing Sheep in Organic Vineyards: An On-Farm Study about Risk of Chronic Copper Poisoning"

_sustainability, doi:10.3390/su132212860_

Round 1

Reviewer 1 Report

Dear authors,

After I read your manuscript, I recommend rejection.

The overall idea of your manuscript is novel. As a livestock specialist, I could give comments on the animal nutrition bit.

The toxicity of Cu to sheep depends mainly on the ingestion rate which depends on: (1) level of Cu in food and (2) food intake by sheep. Your study did not provide any data regarding hours of grazing, leaves amount in the yard or dry matter intake of leaves (or weeds). Accordingly, your results cannot be applied to other scenarios with yards having different leaves production. The recommendation of the study is so vague. The study should instead give a specific recommendation about the dry matter intake of vine leaves per sheep (expressed as g/kg live weight) which is safe.

Please, accept my appologies for my decision.

Author Response

Please find attached our response to your comments.

Reviewer 2 Report

This work aimed to evaluate the effects of sheep grazing on vineyards  and its effects on animal's health, with emphasys on Copper Chronic Poisoning. The work was very well designed and conducted, with  results clearly presented and discussed, and conclusions supported by the results. I just suggested few adjustments, specially on formal aspects. Please see my comments on attached PDF.

Author Response

(The authors gave the same response as above.)

Reviewer 3 Report

I do enjoy reading and evaluating this manuscript. its well written and well designed study. However, there are few concerns about this work.

  1. The number of used animals is low (half of them got some problems), so I believe with this number we can not draw a very strong recommendation, please clarify
  2. what the authors think will happen to the animals if they extended the period of grazing?
  3. when is the best time for the producers to take out the animals form the pastures with the same conditions? Do they have some indicators that can be followed?
  4. what are the other issues that must be considered when grazing these pastures? 

Author Response

(The authors gave the same response as above.)

Reviewer 4 Report

The topic of your manuscript has an exceptional practical and scientific contribution and is very current. I did not notice any shortcomings in the paper that would need to be corrected. 

Author Response

(The authors gave the same response as above.)

Reviewer 5 Report

Introduction

Please, described clearly the hypothesis and the objectives of the study.

The objectives of the study should be inserted also at the start of the Abstract.

Materials & Methods

2.2. Please note that 12 animals do not constitute a flock, hence please rephrase the title.

Please present the reproductive stage of the ewes.

It is not clear for how long the animals grazed each plot. Please describe accurately.

2.4. Data treatment is not good English, hence please rephrase.

Why no liver biopsies were taken?

Please write all the section in a past tense.

Results

Tables are placed at wrong locations within the text.

How were the data for precipitation obtained? This was not described in M&M. Please explain.

Having Results and Discussion in one section is confusing. Please separate. The discussion will be fully evaluated at the second round of review process.

All in all, the manuscript can advance but needs significant improvement as indicated.

Re-evaluation is necessary.

Author Response

(The authors gave the same response as above.)

Round 2

Reviewer 1 Report

Dear Authors,

Many thanks for the clarifications.

However, the copper consumed by sheep , which is my major major concern, is still not presented.

Reviewer 3 Report

Authors have addressed all my comments. I believe the manuscript is ready for publication. Thanks

Author Response

Many thanks for your reviewing work.

Best regards,

Reviewer 5 Report

The authors have improved the manuscript and have responded to the comments in a satisfactory manner (although not perfectly).

Points for correction:
nulliparous empty ewes = ewe-lambs (empty is term used by farmers, not by scientists....), please rephrase.
Tables and Figures should be placed immediately after the paragraph in which they are first referred to, please look into the instructions for authors and correct.
Results and Discussion in the same section is confusing. I understand that the authors do not wish to do, so the editor can decide.

Author Response

Dear Reviewer,

Thank you for your useful comments about our manuscript. The position of tables and figures have been modified according to your advice, as has been the designation of the animals.

We understand that you may find confusing the fact that results and discussion are embedded. As explained before, this complex, trans-disciplinary subject has led us to do this choice, and we hope that it will help readers to better understand our experimental work and interpretation. Your advice to leave the decision to the Editor seems very wise, and should the Editor decide that this aspect has to be modified, we will obviously comply with this demand.

Best regards,

Round 3

Reviewer 1 Report

Dear authors,

You did not provide data on feed intake of sheep. Thus, I still recommend rejection.

Cheers